SciPost Physics

Submission

# Forecasting Generative Amplification

Henning Bahl[1], Sascha Diefenbacher[2], Nina Elmer[1], Tilman Plehn[1,3], and Jonas Spinner[1]

**1** Institut für Theoretische Physik, Universität Heidelberg, Germany
**2** Physics Division, Lawrence Berkeley National Laboratory, Berkeley, USA
**3** Interdisciplinary Center for Scientific Computing (IWR), Universität Heidelberg, Germany

October 17, 2025

## Abstract

Generative networks are perfect tools to enhance the speed and precision of LHC simulations. Especially when generating events beyond the size of the training dataset, it is important to understand their statistical precision. We present two complementary methods to estimate the amplification factor without large holdout datasets. Averaging amplification uses Bayesian networks or ensembling to estimate amplification from the precision of integrals over given phase-space volumes. Differential amplification uses hypothesis testing to quantify amplification without any resolution loss. Applied to state-of-the-art event generators, both methods indicate that amplification is already possible in specific regions of phase space.

# 1 Introduction

The enormous amount of data at the Large Hadron Collider (LHC) allows us to explore the Standard Model with unprecedented precision. Over the next decade, the High Luminosity LHC (HL-LHC) will increase the amount of data by another order of magnitude, promising a significant jump in precision. To analyze this data will require a corresponding increase in the precision and the amount of simulated data.

LHC simulations rely on a comprehensive Monte Carlo event generation and detector simulation chain. It starts with the interaction of partons, from where the events are propagated through jet radiation and hadronization steps to the detector simulation. All steps are based on first principles, with a high level of precision. This link to proper physics models comes with significant computational costs. As a result, the need for HL-LHC simulations exceeds our computational budget by a significant factor.

To solve this pressing problem, we need more efficient simulation methods. Modern machine learning (ML) provides such methods [1, 2], including generative networks that learn the underlying phase space densities from a controlled dataset. For instance, we can train a generative network on classical simulation and use it to generate extra simulation data significantly faster. Generative ultra-fast simulation rests on the assumption that samples drawn from the network can exceed the statistical limitations of the training data, the so-called *amplification**. It has been shown to work for synthetic data [3–5] and detector simulations [6]. This amplification behavior goes back to the interpolation of the underlying density given the inductive bias of the generative network.

Potentially amplifying generative networks have been developed for phase-space sampling [7–14], end-to-end event generation [15–22], hadronization [23, 24], and detector simulators trained on full simulations [25–50]. Generative amplification is complemented with learned smooth amplitude surrogates [21, 51–62] and can be systematically benchmarked using trained classifiers [63]. To quantify the amount of amplification in these different applications, we have, until now, relied on knowing the true underlying distribution or on a large holdout dataset [3, 4, 6]. In actual physics applications with limited statistics, these approaches are not practical. In this work, we investigate methods to quantify amplification factors without knowledge of the true distribution. This way, we address two common concerns: first, knowing the amplification factor of a generative network provides an insight how to use the network for simulations; second, the amplification factor of a generative network provides a lower limit on the statistical uncertainty of a generated dataset. In that sense, reliable amplification is a vital part of generative uncertainty quantification.

The paper is structured as follows: In Section 2 we introduce the fundamental concepts. In Section 3 and Section 4, we present results on the experimental setup for averaging and differential amplification, respectively. Section 5 shows how our amplification estimates can be applied to state-of-the-art generative networks for top pair events at the LHC. Section 6 presents our conclusions.

---

*Earlier we have referred to this effect as GANplification. In this paper we will, strictly speaking, show that L-GATrfication and LLoCa-Transformerification are much more successful. We therefore decide to retire the specific portmanteau.

## 2  Basics and toy datasets

In this section, we very briefly review the idea of generative amplification and present our main toy dataset and network setup. All details of the physics application are given in Section 5.

**Amplification**

Amplification means that a generated dataset provides a better description of the true underlying distribution than the training data. Let $p_{\text{true}}(x)$ be the exact density the generative network should learn. It is represented by a dataset $D_{\text{true}}$ with $n_{\text{train}}$ training events. From this dataset, the generative network extracts $p_{\text{gen}}(x)$,

$$D_{\text{true}}^{n_{\text{train}}} \sim p_{\text{true}}(x) \approx p_{\text{gen}}(x) \qquad \text{with} \qquad |D_{\text{true}}^{n_{\text{train}}}| = n_{\text{train}} \,. \tag{1}$$

Amplification appears if a large generated dataset from the learned density $p_{\text{gen}}(x)$ follows the true density rather than the statistically limited training data

$$\lim_{n_{\text{gen}} \gg n_{\text{train}}} D_{\text{gen}}^{n_{\text{gen}}} \sim p_{\text{gen}}(x) \overset{?}{\sim} p_{\text{true}}(x) \,. \tag{2}$$

Two effects determine the success of amplification:

1. no training is perfect, so $p_{\text{gen}}$ will not reproduce $D_{\text{true}}^{n_{\text{train}}}$ exactly, but in what direction?
2. $p_{\text{gen}}$ will smooth out $D_{\text{true}}^{n_{\text{train}}}$, potentially giving a better approximation to $p_{\text{true}}(x)$.

To quantitatively evaluate Eq. (2), we need a metric to measure the agreement of a dataset $D$ with a probability distribution $p(x)$. Given such a metric $M[D, p(x)]$, we implicitly define the effective number of events $n_{\text{equiv}}$ from the true distribution, such that it approximates $p_{\text{true}}$ as well as an infinitely sampled dataset from the learned $p_{\text{gen}}$,

$$M\left[D_{\text{true}}^{n_{\text{equiv}}}, p_{\text{true}}\right] \overset{!}{=} \lim_{n_{\text{gen}} \to \infty} M\left[D_{\text{gen}}^{n_{\text{gen}}}, p_{\text{true}}\right] \,. \tag{3}$$

Because $p_{\text{gen}}$ and therefore $D_{\text{gen}}$ implicitly depend on the training statistics $n_{\text{train}}$, this relation defines the amplification factor

$$G = \frac{n_{\text{equiv}}}{n_{\text{train}}} \,. \tag{4}$$

In realistic applications, we do not have access to $p_{\text{true}}(x)$, so we can only rely on scaling properties of $M[G, p]$.

In this paper, we aim to estimate $G$ without a large holdout dataset using two different approaches to define the metric $M[D, p(x)]$, *averaging* amplification and *differential* amplification. Both methods are designed to not require information about the true distribution. When the true distribution is known analytically or implicitly available as a large holdout dataset, we use it to validate our two amplification estimates. We start with a toy setup with a simple data distribution and an interpretable generative network. After that, we apply both methods to a proper LHC simulation problem.

**Toy dataset and network**

As a toy dataset, we use a $d$-dimensional Gaussian ring distribution, with radial distribution

$$p_R(x) = \mathcal{N}(R; \mu, \sigma^2) \qquad \text{with} \qquad \mu = 1 \,, \qquad \sigma = 0.1 \,, \tag{5}$$

and flat angular distributions $[p_{\phi,1}(x), ..., p_{\phi,d-1}(x)]$. The networks are trained on Cartesian representation $[x_1, ..., x_d]$, with $\Sigma_i x_i^2 = R^2$. We evaluate them in polar coordinates, to ensure that the network learns all correlations and recovers the correct $R$.

We parametrize the density using an autoregressive transformer [19]. It provides a combination of fast sampling with access to exact densities. We approximate the multi-dimensional phase space distribution as an autoregressive product of conditional probabilities

$$p(x_1, \ldots x_n) = p(x_1) p(x_2|x_1) \cdots p(x_n|x_1 \ldots x_{n-1}). \tag{6}$$

The 1D-conditional probabilities $p(x_i|x_1 \ldots x_{i-1})$ are parametrized in terms of a Gaussian mixture

$$p(x_i|x_1 \ldots x_{i-1}) = \sum_{j=1}^{N} w_j \mathcal{N}(x_i|\mu_j, \sigma_j). \tag{7}$$

The parameters $w_j, \mu_j, \sigma_j$ implicitly depend on the condition $(x_1 \ldots x_{i-1})$, and the weights $w_j$ sum to one. Given this analytic density, we can sample it sequentially. We use a log-likelihood loss

$$\mathcal{L} = \langle -\log p(x) \rangle_{x \sim p(x)} = \left\langle -\sum_{i=1}^{n} \log p(x_i|x_1 \ldots x_{i-1}) \right\rangle_{x \sim p(x)}. \tag{8}$$

In each autoregressive step, we use a transformer architecture to predict $(w, \mu, \sigma)_i$ characterizing $p(x_i|x_1 \ldots x_{i-1})$ conditioned on the prior phase space components $x_1 \ldots x_{i-1}$. Specifically, we embed each component $x_k$ as a transformer token and extract a set of Gaussian mixture parameters from each token. The first set of mixture parameters $(w, \mu, \sigma)_1$ is predicted based on a dummy start token $x_0 = 0$.

## 3 Averaging amplification factor

One way to quantify the agreement between the dataset $D_{\text{gen}}^{n_{\text{gen}}}$ and the probability density $p_{\text{true}}(x)$ in Eq. (3), without knowing this distribution explicitly, is by means of an averaging amplification factor. It uses integrals over phase space volumes $V$ with a reference value

$$I(p_{\text{true}}) = \int_V dx \, p_{\text{true}}(x), \tag{9}$$

generalizing previous approaches [4, 64]. We can compare the true integral to the fraction of points of $D_{\text{gen}}^{n_{\text{gen}}}$ which lie in $V$,

$$\bar{I}(D_{\text{gen}}^{n_{\text{gen}}}) = \frac{1}{n} \sum_{x \in D_{\text{gen}}^{n_{\text{gen}}},} \mathbf{1}_{x \in V} \qquad \text{with} \qquad \mathbf{1}_{x \in V} = \begin{cases} 1 & x \in V \\ 0 & \text{else} \end{cases}. \tag{10}$$

### 3.1 Metric extrapolation and Bayesian networks

The deviation between the true density and this sample over $V$ can then be measured as

$$M_I\left[D_{\text{gen}}^{n_{\text{gen}}}, p_{\text{true}}\right] = \left\langle \left[I(p_{\text{true}}) - \bar{I}(D_{\text{gen}}^{n_{\text{gen}}})\right]^2 \right\rangle$$

$$= \begin{cases} \sigma_{\text{stat}}^2(n_{\text{gen}}) & p_{\text{gen}} = p_{\text{true}} \\ \sigma_{\text{stat}}^2(n_{\text{gen}}) + \sigma_{\text{model}}^2(p_{\text{gen}}, p_{\text{true}}) & p_{\text{gen}} \neq p_{\text{true}} \end{cases}. \tag{11}$$

The uncertainty due to mis-modeling $p_{\text{true}}$ as $p_{\text{gen}}$ is $\sigma_{\text{model}}$. The statistical uncertainty $\sigma_{\text{stat}}$ is non-zero even when $p_{\text{gen}}(x) = p_{\text{true}}(x)$, and its scaling is described by classical statistics. It follows a binomial distribution and can therefore be extrapolated. Using the standard deviation of the binomial distribution, we find

$$M_I\left[D_{\text{true}}^{n_{\text{equiv}}}, p_{\text{true}}\right] = \sigma_{\text{stat}}^2(n_{\text{equiv}}) \simeq \frac{\bar{I}(D_{\text{true}}^{n_{\text{train}}})[1 - \bar{I}(D_{\text{true}}^{n_{\text{train}}})]}{n_{\text{equiv}}}$$

$$\lim_{n_{\text{gen}} \to \infty} M_I\left[D_{\text{gen}}^{n_{\text{gen}}}, p_{\text{true}}\right] = \sigma_{\text{model}}^2(p_{\text{gen}}, p_{\text{true}}) \neq 0 \,. \tag{12}$$

The next step towards evaluating Eq. (3) is to determine $\lim M_I[D_{\text{gen}}^{n_{\text{gen}}}, p_{\text{true}}]$. One strategy uses Bayesian Neural Networks (BNNs) [65–68], specifically Bayesian generative networks [19,69, 70]. Their learned uncertainty should exactly describe the difference between $p_{\text{gen}}$ and $p_{\text{true}}$. An alternative which we employ for our physics application is (repulsive) ensembles [62,71, 72]. BNNs approximate the posterior of the neural network parameters given by the training dataset with a variational inference ansatz,

$$p(\theta|D_{\text{train}}) \approx q(\theta) \,. \tag{13}$$

We can express the density learned by the Bayesian generative network as conditional on the network parameters, $p_{\text{gen}}(x|\theta)$. For the evaluation, we sample $N_{\text{BNN}}$ sets of network parameters $\theta_1, \ldots \theta_{N_{\text{BNN}}}$ from $q(\theta)$. For each parameter set, we sample a generated dataset $D_{\text{gen},\theta_i}^{n_{\text{gen}}}$ and define the fraction of events in $V$ as defined in Eq. (10) and its expectation value over the $\theta_i$ as

$$\bar{I}_{\theta_i} = \frac{1}{n_{\text{gen}}} \sum_{x \in D_{\text{gen},\theta_i}^{n_{\text{gen}}}} \mathbf{1}_{x \in V} \qquad \Rightarrow \qquad \langle \bar{I} \rangle_\theta = \frac{1}{N_{\text{BNN}}} \sum_i \bar{I}_{\theta_i} \,. \tag{14}$$

The corresponding variance should give our difference measure

$$M_I\left[D_{\text{gen}}^{n_{\text{gen}}}, p_{\text{true}}\right] = \langle \bar{I}^2 \rangle_\theta - \langle \bar{I} \rangle_\theta^2$$

$$= \sigma_{\text{model}}^2(p_{\text{gen}}, p_{\text{true}}) + \sigma_{\text{stat}}^2(n_{\text{gen}}) \,. \tag{15}$$

A simple statistical scaling gives us, just like in Eq. (12),

$$\sigma_{\text{stat}}(n_{\text{gen}}) = \frac{\langle \bar{I} \rangle_\theta (1 - \langle \bar{I} \rangle_\theta)}{n_{\text{gen}}} \qquad \text{and} \qquad \sigma_{\text{stat}}(n_{\text{equiv}}) = \frac{\langle \bar{I} \rangle_\theta (1 - \langle \bar{I} \rangle_\theta)}{n_{\text{equiv}}} \tag{16}$$

This scaling allows us to evaluate the model uncertainty in Eq. (3)

$$\lim_{n_{\text{gen}} \to \infty} M_I\left[D_{\text{gen}}^{n_{\text{gen}}}, p_{\text{true}}\right] = \sigma_{\text{model}}^2(p_{\text{gen}}, p_{\text{true}}) = \langle \bar{I}^2 \rangle_\theta - \langle \bar{I} \rangle_\theta^2 - \frac{\langle \bar{I} \rangle_\theta (1 - \langle \bar{I} \rangle_\theta)}{n_{\text{gen}}} \,. \tag{17}$$

It turns the definition of $n_{\text{equiv}}$ from Eq. (3) into

$$M_I\left[D_{\text{true}}^{n_{\text{equiv}}}, p_{\text{true}}\right] \simeq \frac{\langle \bar{I} \rangle_\theta (1 - \langle \bar{I} \rangle_\theta)}{n_{\text{equiv}}} \overset{!}{=} \langle \bar{I}^2 \rangle_\theta - \langle \bar{I} \rangle_\theta^2 - \frac{\langle \bar{I} \rangle_\theta (1 - \langle \bar{I} \rangle_\theta)}{n_{\text{gen}}} \,, \tag{18}$$

extrapolated from finite $n_{\text{gen}}$. Our derivation relies on the scaling of $M[D, p]$ to (i) extrapolate into the limit $n_{\text{gen}} \to \infty$, and (ii) trace the dependence on $n_{\text{equiv}}$.

Once we know how to estimate $\sigma_{\text{model}}$, we can extract the maximum number of meaningfully generated events, $n_{\text{equiv}}$, and the amplification factor. This procedure is visualized in

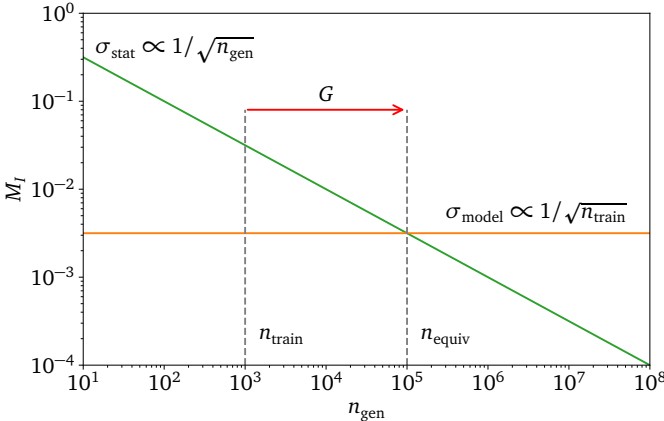

Figure 1: Illustration of the averaging amplification estimate. The statistical uncertainty of the generated dataset is shown in green; the model uncertainty of the generative network is shown in orange.

Fig. 1. The model uncertainty on the right-hand side of Eq. (3) is estimated by the BNN and typically scales as $1/\sqrt{n_{\text{train}}}$. For the statistical uncertainty on the left-hand side of Fig. 1 we use the scaling of $\sigma_{\text{stat}}$ as $1/\sqrt{n_{\text{gen}}}$. The crossing of the lines determines $n_{\text{equiv}}$ and therefore $G = n_{\text{equiv}}/n_{\text{train}}$.

This averaging amplification factor will depend strongly on the choice of integration region $V$. A very small integration region will not contain enough events, while using the entire phase space leads to a meaningless result, as $I = 1 = \bar{I}$ by definition. We can split the phase space into $n_V$ non-overlapping regions $V_i$ with $i \in [0, n_V]$ and obtain the combined uncertainty over all $V_i$ as a quadratic sum,

$$\sigma_{\text{stat}}^2 = \sum_i \sigma_{\text{stat},V_i}^2 \qquad \text{and} \qquad \sigma_{\text{model}}^2 = \sum_i \sigma_{\text{model},V_i}^2 \ . \tag{19}$$

Choosing $V_i$ as quantile regions links this method to the approach of Ref. [3].

## 3.2 1D Gaussian fit illustration

As a first illustration of Fig. 1, we look at a 1D unit Gaussian as the true distribution

$$p_{\text{true}}(x) = \mathcal{N}(x; 0, 1) \,, \tag{20}$$

In place of a generative network, we use a unit-width Gaussian with variable mean $\hat{\mu}$ to approximate the true distribution. A maximum likelihood fit is equivalent to calculating the mean and standard deviation from the training data. The distribution of $\hat{\mu}$ across many fits follows a Gaussian with width $\sigma_{\hat{\mu}}$,

$$\hat{\mu} \sim \mathcal{N}(\hat{\mu}; 0, \sigma_{\hat{\mu}}) \qquad \text{with} \qquad \sigma_{\hat{\mu}} = \frac{1}{\sqrt{n_{\text{train}}}} \ . \tag{21}$$

Using this fitted mean $\hat{\mu}$, we draw $n_{\text{gen}}$ points from $\mathcal{N}(x; \hat{\mu}, 1)$, just as from a generative network

$$D_{\text{gen}}^{n_{\text{gen}}} \sim p_{\text{gen}}(x) = \mathcal{N}(x; \hat{\mu}, 1) \,. \tag{22}$$

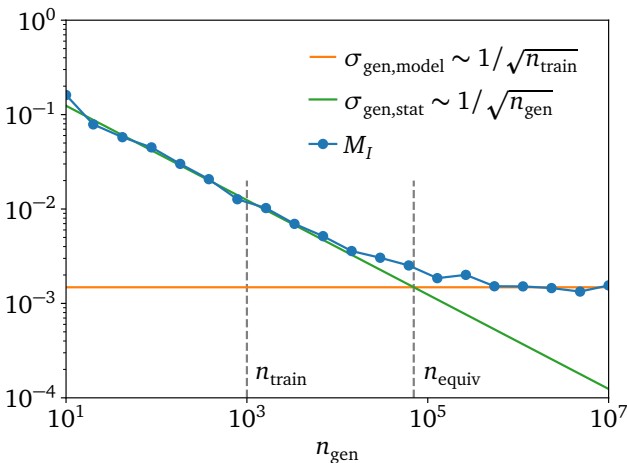

Figure 2: Scaling of the mean quadratic deviation from the true integral value as a function of the number of samples drawn from the fitted Gaussian. The curve points are averaged over 50 independent experiments. **replace $\sigma_{\text{true}}$ in all plots.**

For the averaging amplification estimate we calculate the average amplification factor over a phase space interval $V = [a, b]$ using the integral

$$I = \int_a^b dx \, \mathcal{N}(x; 0, 1) = \frac{1}{2}\left[\text{erf}\left(\frac{b}{\sqrt{2}}\right) - \text{erf}\left(\frac{a}{\sqrt{2}}\right)\right]. \tag{23}$$

Next, we compute the fraction of points in $D_{\text{gen}}^{n_{\text{gen}}}$ within $V$. Given that the number of points in $[a, b]$ follows a binomial distribution, the statistical uncertainty from the finite number of points can be approximated as in Eq. (12). For a small target region $V$ and $I \ll 1$ the left-hand side of Eq. (3) can then be approximately written as

$$M_I\left[D_{\text{true}}^{n_{\text{equiv}}}, p_{\text{true}}\right] = \sigma_{\text{stat}}^2(n_{\text{equiv}}) \simeq \frac{I(1-I)}{n_{\text{equiv}}} \simeq \frac{I}{n_{\text{equiv}}}. \tag{24}$$

The second ingredient of our amplification estimate is the model uncertainty, in our case from the imperfect fit value $\hat{\mu}$. The exact integral over $p_{\text{gen}}(x) = \mathcal{N}(x; \hat{\mu}, 1)$ is

$$\begin{aligned}
I' &= \frac{1}{2}\left[\text{erf}\left(\frac{b - \hat{\mu}}{\sqrt{2}}\right) - \text{erf}\left(\frac{a - \hat{\mu}}{\sqrt{2}}\right)\right] \\
&= I + \frac{\hat{\mu}}{\sqrt{2\pi}}\left(e^{-a^2/2} - e^{-b^2/2}\right) + \mathcal{O}(\hat{\mu}^2).
\end{aligned} \tag{25}$$

The uncertainty of $\hat{\mu}$ is given in Eq. (21), so we can estimate the model uncertainty using error propagation,

$$\sigma_{\text{model}}(n_{\text{train}}) = \left|\frac{dI'}{d\hat{\mu}}\right| \sigma_{\hat{\mu}} = \frac{1}{\sqrt{2\pi}}\left|e^{-a^2/2} - e^{-b^2/2}\right| \frac{1}{\sqrt{n_{\text{train}}}}. \tag{26}$$

The total generative uncertainty as a function of $n_{\text{gen}}$ is then given by

$$M_I\left[D_{\text{gen}}^{n_{\text{gen}}}, p_{\text{true}}\right] = \sigma_{\text{stat}}^2(n_{\text{gen}}) + \sigma_{\text{model}}^2(n_{\text{train}}) \simeq \frac{I}{n_{\text{gen}}} + \left|\frac{dI'}{d\hat{\mu}}\right|^2 \frac{1}{n_{\text{train}}}$$

$$\Rightarrow \qquad \lim_{n_{\text{gen}} \to \infty} M_I\left[D_{\text{gen}}^{n_{\text{gen}}}, p_{\text{true}}\right] = \sigma_{\text{model}}^2(n_{\text{train}}) \simeq \left|\frac{dI'}{d\hat{\mu}}\right|^2 \frac{1}{n_{\text{train}}}. \tag{27}$$

This gives us for the implicit definition of $n_{\text{equiv}}$ in Eq. (3)

$$\sigma^2_{\text{stat}}(n_{\text{equiv}}) = \frac{I}{n_{\text{equiv}}} \stackrel{!}{=} \sigma^2_{\text{model}}(n_{\text{train}}) = \left| \frac{dI'}{d\hat{\mu}} \right|^2 \frac{1}{n_{\text{train}}} \ . \tag{28}$$

In the left panel of Fig.2 we compute this case for two Gaussians, with the interval $[0, 0.5]$ and from $n_{\text{train}} = 1000$ training events. This gives us for the two uncertainties

$$\sigma_{\text{stat}}(n_{\text{gen}}) = \sqrt{\frac{0.154}{n_{\text{gen}}}} \qquad \text{and} \qquad \sigma_{\text{model}}(n_{\text{train}}) = \sqrt{\frac{0.002}{n_{\text{train}}}} \ . \tag{29}$$

and in turn

$$n_{\text{equiv}} \simeq \frac{0.154}{0.002} n_{\text{train}} = 70 \, n_{\text{train}} \qquad \text{and} \qquad G = \frac{n_{\text{equiv}}}{n_{\text{train}}} \simeq 70 \ . \tag{30}$$

The numerical results for the two uncertainties in Fig. 2, averaged over 50 independent experiments, agree well with the analytic prediction. We can also extract $n_{\text{equiv}}$ graphically from the crossing of the two contributions, confirming $G = 70$.

## 3.3 Gaussian-ring toy data

To also validate the extracted amplification factor using BNNs for the model uncertainty, we use Gaussian ring toy data in two and four dimensions, as described in Section 2. We divide the phase space along the radius direction $R$ into 20 regions, each containing 5% of the events, as shown in Fig. 3. We label the regions sequentially from left to right.

For each region, we estimate the integral in Eq. (14) from samples generated with the Bayesian autoregressive transformer and compare it to the true integral values. The resulting amplification curves are shown in Fig. 4. Since $\sigma_{\text{model}}$ is calculated via subtraction, it can become negative due to numerical fluctuations. We do not display these points. For region 2, shown in the upper panels of Fig. 4, the curves displaying the true and estimated uncertainties $\sigma_{\text{true}}$ overlap, so the BNN learns the uncertainties correctly. The extracted $\sigma_{\text{stat}}$ and $\sigma_{\text{model}}$ also follow the expected scaling behaviors. Consequently, the estimated and true amplification factors agree well with $G = 2.6$ and $G = 1.9$ for the two dimensionalities.

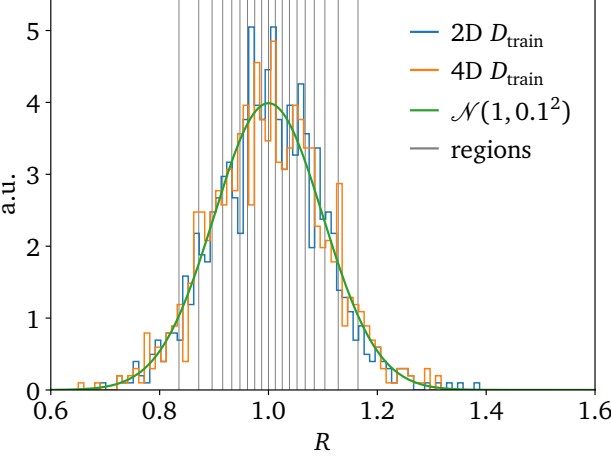

Figure 3: Training data and integral regions for Gaussian ring.

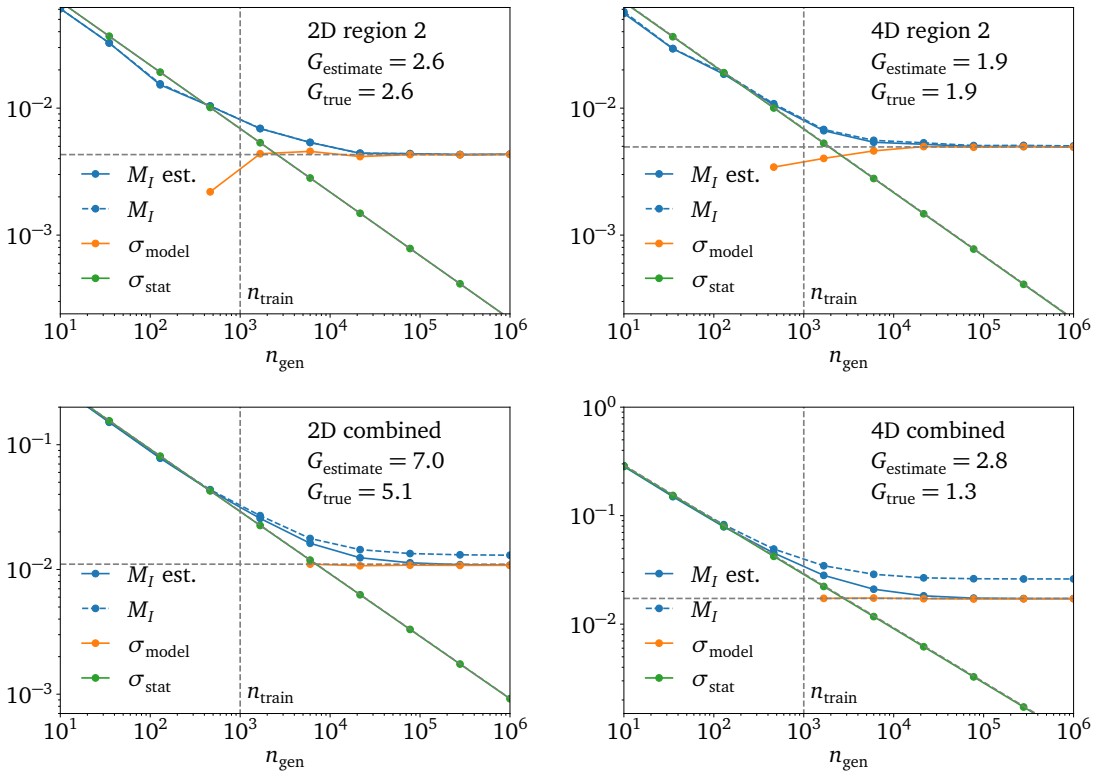

Figure 4: Actual standard deviation and estimated uncertainties for various radial integrals over a Gaussian ring. Left: 2D Gaussian ring. Right: 4D Gaussian ring.

Next, we combine all regions except for the first and the last in the lower row of Fig. 5. The combination follows Eq. (19), and the estimated $\sigma_{\text{true}}$ is slightly smaller than the true uncertainty. Still, the estimated amplification factors agree reasonably well with the true factors. In the 2D case, the amplification factor is $G \approx 7$, in the more challenging 4D case, we find $G \approx 2.8$, starting to deviate from the true value $G \approx 1.3$.

Finally, we study region 20, the tail of the radial distribution. The estimated $M_I$ again tracks the true $\sigma_{\text{true}}$, but the agreement is worse than in region 2. The predicted amplification factors in the tail region are smaller than one for two dimensions and close to zero for four

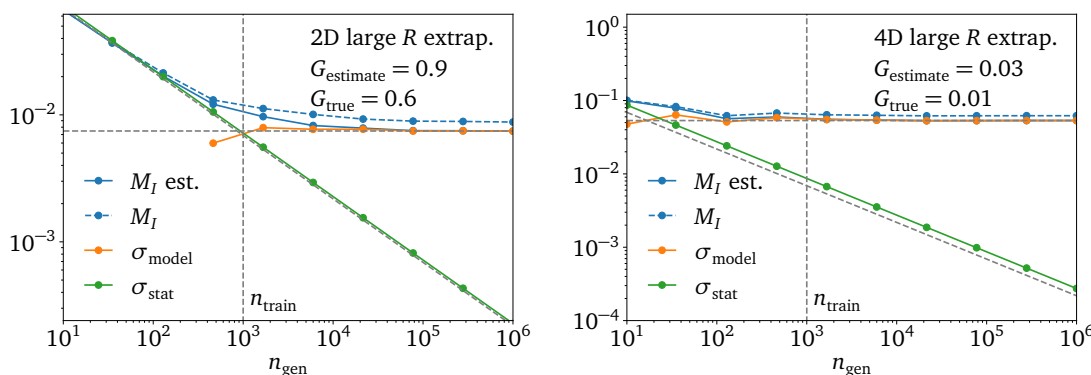

Figure 5: Actual standard deviation and estimated uncertainties for radial integrals, including only regions with large radius $R$. Left: 2D Gaussian ring. Right: 4D Gaussian ring.

dimensions. This means the generative network does not learn the underlying density correctly in the low-statistics tails. Crucially, our averaging approach also predicts this amplification failure correctly.

# 4 Differential amplification factor

Extracting the (averaging) amplification factor from the integral $I$ implies that our estimate is local and that it does not capture any features below the resolution of $I$. This motivated an alternative, differential amplification factor. First, we turn Eq. (3) from a definition of $n_{\text{equiv}}$ in terms of one dataset and one density into a comparison of two datasets,

$$M\left[D_{\text{gen}}^{n_{\text{gen}}}, p_{\text{true}}\right] \to M\left[D_{\text{gen}}^{n_{\text{gen}}}, D_{\text{true}}^{n_{\text{holdout}}}\right] \tag{31}$$

For now, we assume that $p_{\text{true}}$ is available through a large holdout dataset. The implicit definition of $n_{\text{equiv}}$ in Eq. (3) now reads

$$M\left[D_{\text{true}}^{n_{\text{equiv}}}, D_{\text{true}}^{n_{\text{holdout}}}\right] \overset{!}{=} \lim_{n_{\text{gen}} \to \infty} M\left[D_{\text{gen}}^{n_{\text{gen}}}, D_{\text{true}}^{n_{\text{holdout}}}\right]. \tag{32}$$

Because we do not have a large holdout dataset we need a metric with a known asymptotic to extrapolate from $n_{\text{train}}$.

This approach requires a two-sample test statistic $M$ for which the left-hand side of Eq. (32) is known. This is the case if $M$ has an analytically known asymptotic behavior for both samples drawn from identical distributions. This requirement limits us to uni-variate test statistics. While multivariate test statistics exist, e.g. based on maximum mean discrepancy, energy distance, or the earth mover's distance, none of them have a known asymptotic behavior. Further, several uni-variate test statistics, such as the $\chi^2$ test, Student-t test, and F-test, assume a certain form of the 1D distribution, which we can not guarantee for an arbitrary generative task. Another set of methods, for example the Mann-Whitney U test, only test for differences in the mean or variance of the distributions, but not for the general distribution equality that we want to test. This leaves us with the Kolmogorov-Smirnov (KS) test [73, 74], which is cheap to evaluate and has a known asymptotic behavior.

To perform uni-variate tests, we need a summary statistic that compresses the high-dimensional phase space information into a 1D variable, which is designed to discriminate between the two datasets. For specific applications, a physically relevant feature might be the best choice. As a general approach, we train a classifier to separate a generated set from the training set. The output of the trained classifier approximates the likelihood ratio for each point in any dataset, describing its agreement with the training data. This likelihood ratio is the most powerful summary statistic according to the Neyman-Pearson lemma, satisfying our criteria for a summary statistic.

## 4.1 Kolmogorov-Smirnov test

The KS distance or test statistic between two 1D datasets $D_1 \sim p_1(x)$ and $D_2 \sim p_1(x)$ is defined in terms of their empirical cumulative distribution functions $F$,

$$M_{\text{KS}}[D_1, D_2] = \sup_y |F(y, D_1) - F(y, D_2)|$$

$$\text{with} \quad F(y, D) = \frac{1}{n} \sum_{y_i \in D} \mathbf{1}_{y_i < y} \quad \text{with} \quad |D| = n \quad \text{and} \quad \mathbf{1}_{y_i < y} = \begin{cases} 1 & y_i < y \\ 0 & \text{else} \end{cases}. \tag{33}$$

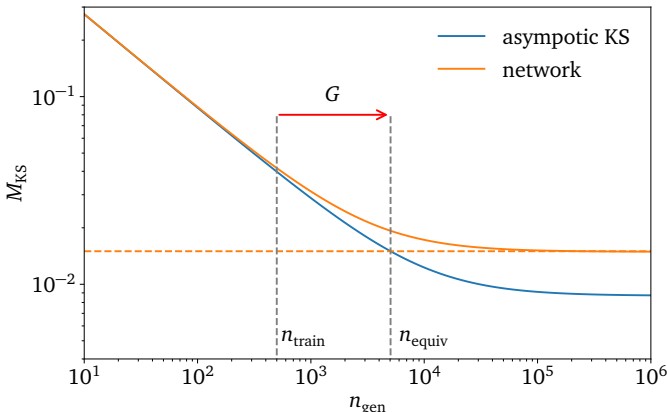

Figure 6: Conceptual sketch of the differential amplification method based on the KS test. The asymptotic behaviour of the KS test is shown in blue; the actual KS test value obtained by comparing the generated and training datasets, in orange.

Crucially, for $p_1(x) = p_2(x)$ the KS test statistic has an asymptotic behavior for large $n_{1,2}$, where the rescaled KS statistic follows the Kolmogorov distribution $p_K(K)$

$$\sqrt{\frac{n_1 n_2}{n_1 + n_2}} M_{\text{KS}}[D_1, D_2] = K \sim p_K(K) . \tag{34}$$

The scaling factor absorbs the two known statistical uncertainties $1/n_1 + 1/n_2$. We are only interested in the finite expectation value of this rescaled KS statistic, which means we can use the expectation value of the Kolmogorov distribution to arrive at

$$\sqrt{\frac{n_1 n_2}{n_1 + n_2}} \langle M_{\text{KS}}[D_1, D_2] \rangle = \langle K \rangle = \sqrt{\frac{\pi}{2}} \log 2 . \tag{35}$$

The best approximation we have for $n_{\text{holdout}}$ is $n_{\text{train}}$. Because the left-hand side of Eq. (32) involves two sets from identical distributions, we can insert this form and extract the dependence on $n_{\text{equiv}}$

$$\sqrt{\frac{\pi}{2}} \log 2 \sqrt{\frac{n_{\text{gen}} + n_{\text{train}}}{n_{\text{gen}} n_{\text{train}}}} \Bigg|_{n_{\text{gen}} = n_{\text{equiv}}} \stackrel{!}{=} \lim_{n_{\text{gen}} \to \infty} \langle M_{\text{KS}}[D_{\text{gen}}^{n_{\text{gen}}}, D_{\text{true}}^{n_{\text{train}}}] \rangle . \tag{36}$$

Based on this formula we illustrate in Fig. 6 how to extract $n_{\text{equiv}}$. First, we calculate the right-hand side as a function of $n_{\text{gen}}$ and determine the limit $n_{\text{gen}} \to \infty$. Then we compute the left-hand side as a function of $n_{\text{equiv}}$ and determine the value $n_{\text{equiv}} = n_{\text{gen}}$ where it crosses the known value for the right-hand side. The figure also shows the weakness of out KS-estimate: the asymptotic KS curve flattens and becomes independent of $n_{\text{equiv}}$ in the limit $n_{\text{equiv}} \gg n_{\text{train}}$.

In addition to the mean of the KS test statistics converging to the rescaled mean of the Kolmogorov distribution, we know that the KS test statistic is itself distributed according to a rescaled Kolmogorov distribution for identical distributions. This allows us to quantify deviations from the asymptotic behavior using the confidence levels of the Kolmogorov distribution. Specifically, the Kolmogorov variable $K$ is contained in the interval

$$\langle K \rangle - c(\alpha) < K < \langle K \rangle + c(\alpha) \quad \text{with} \quad c(\alpha) \simeq \sqrt{\frac{1}{2} \ln \left( \frac{2}{\alpha} \right)} . \tag{37}$$

with a probability $1 - \alpha$. We can assign a confidence interval to the amplification factor by replacing $\langle K \rangle$ with $\langle K \rangle \pm c(\alpha)$.

## 4.2 Gaussian-ring toy data

To estimate differential amplification using the KS test, we again resort to the Gaussian ring toy data introduced in Section 2. We use the 2D and 4D versions. For training, we use $1,000$ points, while the generated datasets contain up to $10^7$ points. The KS test is evaluated for three different summary statistics:

1. $R$: radius of the Gaussian ring;
2. $LR_{class}$: likelihood ratio of a classifier trained to distinguish training and generated data;
3. $LR_{truth}$: true likelihood ratio.

First, we test the consistency of the KS test in Fig. 7 by comparing the training dataset with 1000 points against an independent test dataset of variable size. Both datasets are drawn from the truth distribution, so we expect the KS statistic to follow Eq. (36). This asymptotic behavior is shown as a solid black line. The gray bands indicate the one-sigma to three-sigma intervals of the Kolmogorov distribution. We evaluate the KS test on $LR_{class}$ and $R$, but not on $LR_{truth}$, because it is equal for both samples by construction. We estimate uncertainties on the KS statistic using ten independent Bayesian network samples and classifier trainings. Both curves follow the asymptotic KS curve, so as expected the KS test is unable to distinguish between the datasets.

Next, we investigate the 1D summary statistics on which the KS test is based. In addition to the full datasets we also define cut datasets, where all events with a likelihood ratio outside 0.95...1.05 are removed. This uses likelihoods to refine the generated set, similar to using classifier-based likelihood ratios for generative improvement [75]. The corresponding histograms for the training and test datasets are shown in Fig. 8. The upper panels show the training datasets, and the lower panels the full generated dataset. The training and generated datasets are in good agreement, up to numerical fluctuations.

The empirical CDFs, on which the KS test is based, are shown in Fig. 9. Comparing the CDFs of the training and generated datasets, we see small differences which the KS test is sensitive to. Among the three summary statistics, the smallest difference appears for the observable $R$, where all angular dependencies are ignored. In the left and middle panels, we also see that cutting on the likelihood ratio reduces the differences between the cumulative distributions.

We study differential amplification in Fig. 10. We compute the KS test statistics between

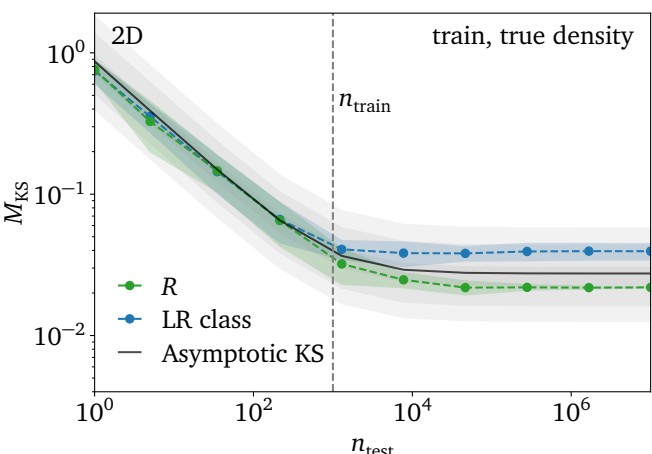

Figure 7: Showing the results for the KS test comparing train vs. an independent test dataset. The uncertainty bands are obtained by averaging over ten classifier samples.

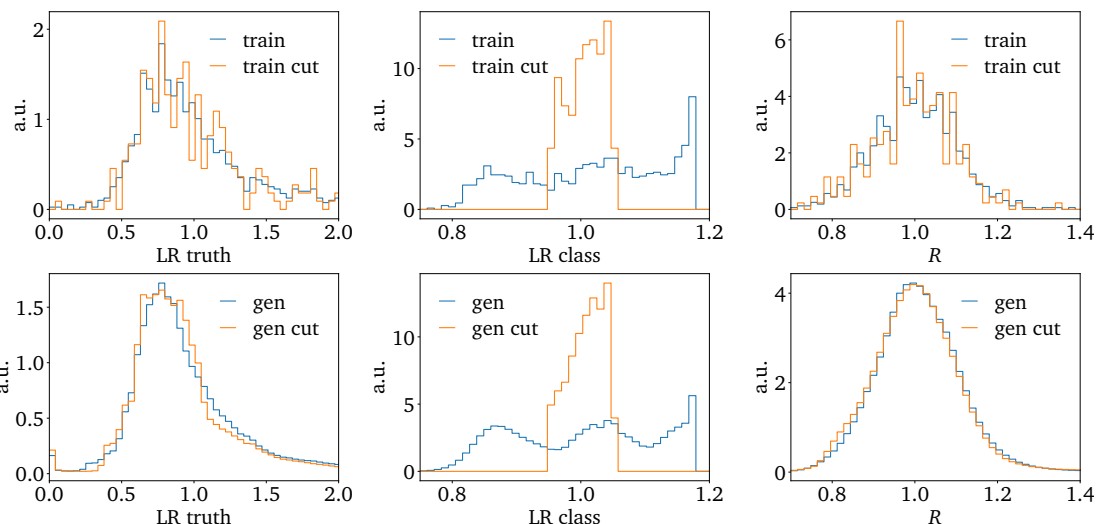

Figure 8: $LR_{truth}$, $LR_{class}$ and $R$ distributions of the training and generated 2D Gaussian ring data datasets. The blue line represents the results for the full dataset, the orange line the results after removing the largest and smallest classifier weights.

the training dataset and differently-sized subsets of the generated data. As in Section 3, we remove the extrapolation regions for small and high $R$ here, as indicated in the plots.

The upper left panel of Fig. 10 shows that the selected summary statistic affects the predicted amplification behavior. Both LR-based summary statistics show no amplification, while we obtain $G \simeq 22$ for the summary statistic $R$. This is explained by the fact that the radius measure is only sensitive to deviations in the radius distribution, while the two classifier scores are sensitive to any discrepancies. Both amplification estimates are valid, but describe different levels of detail.

The upper right panel of Fig. 10 instead compares the generated dataset to a large holdout dataset drawn from the truth distribution. This serves as a cross-check, comparable to Refs. [76, 77], and is not feasible for realistic applications. The amplification behavior is the same as in the left panel. As the large test dataset provides us with a steep asymptotic KS value for large $n_{gen}$, the amplification values can be determined more reliably.

In the lower left panel of Fig. 10, we present the KS test results on the refined generative set, excluding classifier likelihood ratios that deviate significantly from one. The location of these excluded events is shown in the lower right panel of Fig. 10. The most notable effect is that we now observe a higher amplification factor for the classifier-based KS test. This confirms the validity of our likelihood-ratio-based KS test. The effects of the cut on the KS values for

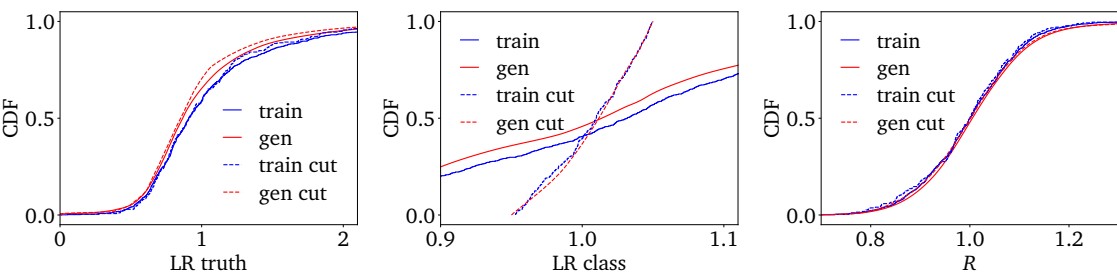

Figure 9: Empirical CDF for $LR_{class}$, $LR_{truth}$, and the radius $R$.

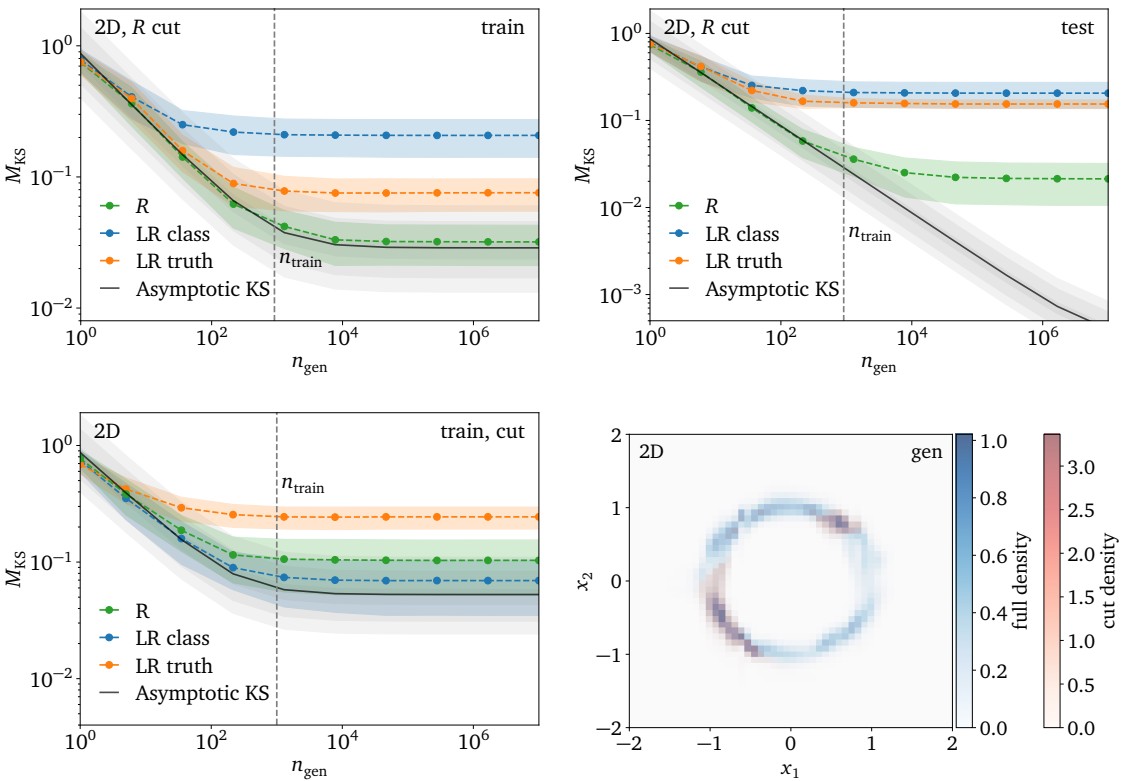

Figure 10: KS test for a 2D Gaussian, with uncertainties based on 50 BNN samples. Upper left: KS test for different 1D summary statistics. Upper right: same, but for a holdout dataset. Lower left: same as upper left, but removing all events with a classifier score outside 0.95...1.05. Lower right: all (blue) and cut points (red) for the generated dataset.

the true likelihood ratio are minimal, due to imperfect training of the classifier.

Finally, we show the results for the standard KS test for the 4D Gaussian ring in Fig. 11. We observe similar trends to those in the 2D case. The radius summary statistic again yields larger

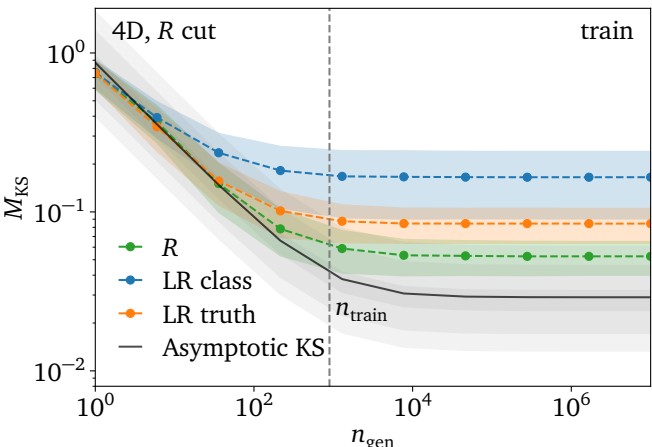

Figure 11: KS test results for a 4D Gaussian toy data. The KS test statistic is shown as a function of $n_{\text{gen}}$ comparing different 1D summary statistics.

amplification factors than the likelihood ratio-based summary statistics. However, the classifier and the true likelihood results now agree within their uncertainties, indicating that the 4D classifier is well trained. The radius summary statistic does not show amplification anymore because learning it precisely in this high-dimensional problem is a challenge for the generative network. These findings are in line with previous tests on higher-dimensional Gaussian rings [3].

# 5 Top pair production

After validating our two ways of estimating amplification on toy datasets, we turn to the generation of reconstruction-level events with two hadronically decaying tops and additional jets.

**Datasets**

We use top pair events [22] from MADGRAPH5_AMC@NLO 3.5.1 [78]. We use PYTHIA8 [79] for the parton shower and hadronization. The detector simulation is performed with DELPHES [80] using the default ATLAS card. The jets are reconstructed using an anti-$k_T$ algorithm with $R = 0.4$ via FASTJET [81]. As phase-space cuts we employ

$$p_{T,j} > 22 \,\text{GeV} \qquad \Delta R_{jj} > 0.5 \qquad |\eta_j| < 5 \qquad N_b = 2 \,. \tag{38}$$

The top quarks are reconstructed using a $\chi^2$ algorithm [82], and all identical particles are sorted by their transverse momentum. We use two datasets with different additional jet multiplicities and the following number of events

$$
\begin{aligned}
t\bar{t} + 0j : & \qquad n_{\text{train}} \sim 5 \cdot 10^5 & \qquad n_{\text{test}} \sim 8 \cdot 10^6 \\
t\bar{t} + 4j : & \qquad n_{\text{train}} \sim 2 \cdot 10^5 & \qquad n_{\text{test}} \sim 2 \cdot 10^5 \,.
\end{aligned}
\tag{39}
$$

**Generative networks**

Instead of the autoregressive generative network used for the toy studies, we use high-performance event generation networks. Conditional flow matching (CFM) generators with the Lorentz-equivariant L-GATr and LLoCa transformers define the state of the art [21,22]. We also employ a vanilla transformer to quantify the gains from Lorentz equivariance.

CFM generators encode phase space densities $p(x_0)$ sampled from a given latent distribution $p_{\text{latent}}(x_1)$. The continuous transition $x(t)$ is described either by an ordinary differential equation (ODE) or by a continuity equation,

$$\frac{dx(t)}{dt} = v(x(t), t) \qquad \text{or} \qquad \frac{\partial \rho(x, t)}{\partial t} = -\nabla_x \left[ v(x(t), t) \, p(x(t), t) \right] , \tag{40}$$

with the same velocity $v(x(t), t)$. The time parameterizes the transition from the latent distribution to the phase space distribution,

$$p(x, t) \to \begin{cases} p(x) & t \to 0 \\ p_{\text{latent}}(x) = \mathcal{N}(x; 0, 1) & t \to 1 \,, \end{cases} \tag{41}$$

The learned velocity field

$$v_\theta(x(t), t) \approx v(x(t), t) \tag{42}$$

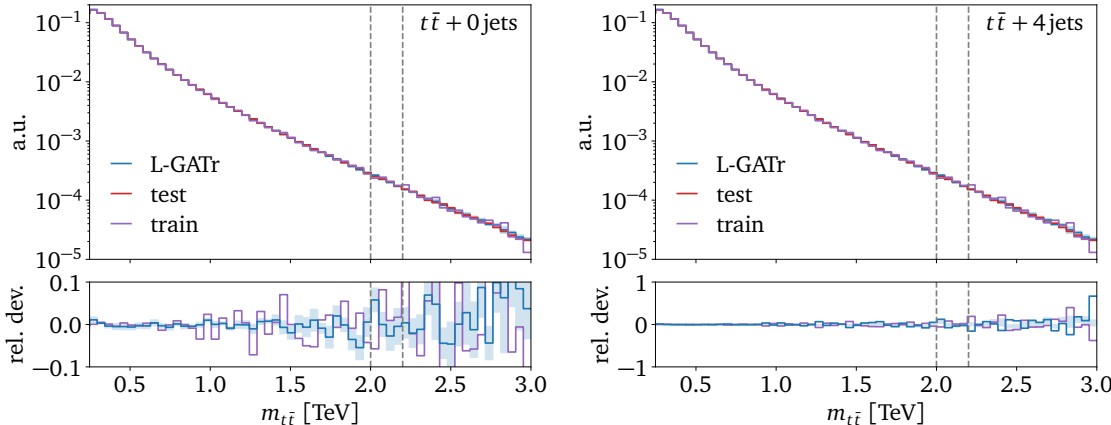

Figure 12: Left: di-top mass from the training and test datasets (upper), and the relative deviation to the test distribution (lower). We also show distributions from the L-GATr CFM network. The colored band indicates the standard deviation around the mean as predicted by the trained NN ensemble. Right: same as left, but for $t\bar{t} + 4\,$jets.

allows us to generate events using an ODE solver.

In principle, any network architecture can be used to encode the velocity field. LHC events at reconstruction level respect a rotational SO(2) symmetry around the beam axis. Embedding it in the velocity network boosts the network performance. The L-GATr [21, 58] and LLoCa transformers [22, 83] implement the larger Lorentz symmetry group. Therefore, we also need to incorporate learned symmetry breaking [21, 22].

To estimate a learned (statistical) uncertainty for the generated data, we use ensembling instead of Bayesian Networks [62, 68, 71, 72, 84, 85]. They include 10 independently trained generative networks for each of the three architectures. While formally a repulsive interaction between the ensemble members is needed to guarantee convergence to the true posterior [84], this repulsive kernel can be neglected for large enough training datasets [62].

## 5.1 Averaging amplification

We start with the averaging amplification estimate, focusing on a specific $m_{t\bar{t}}$ range. The distributions of the training and test datasets are shown in Fig. 12, for $t\bar{t} + 0\,$jets (left) and $t\bar{t} + 4\,$jets (right), compared to the L-GATr generator. We also indicate the chosen phase-space region

$$2\,\text{TeV} \le m_{t\bar{t}} \le 2.2\,\text{TeV} \,. \tag{43}$$

For $t\bar{t} + 0\,$jets production, we see that the test dataset is, as expected, noisier than the training dataset in the tail of the distribution. This is also reflected in the L-GATr results, for which we observe a growing spread of the ensemble with increasing $m_{t\bar{t}}$. The behavior is similar for $t\bar{t} + 4\,$jets production. Since the size of the test dataset is much smaller, we observe increasing noise for smaller $m_{t\bar{t}}$ values. The L-GATr distributions remain stable over the entire $m_{t\bar{t}}$ range.

Following Section 3, we estimate $M_I$ for the generative networks within this phase-space region. For $t\bar{t} + 0\,$jets production, we can predict the true integral to $8 \cdot 10^{-4} \pm 1 \cdot 10^{-5}$, a precision sufficient to validate our averaging amplification results. In Fig. 13, we illustrate the averaging estimate for $t\bar{t} + 0\,$jets in the left panel. All three architectures show good agreement between the estimated and true uncertainties. For the transformer and L-GATr architectures, the agreement is nearly perfect. For the LLoCa transformer, we observe slight

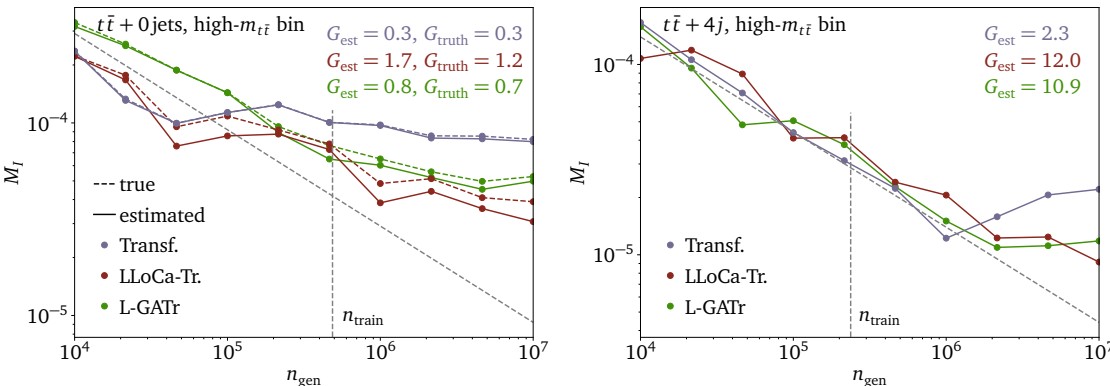

Figure 13: Left: averaging amplification test for $t\bar{t}$ production using three different $t\bar{t}$ event generators. The integral is computed in the region $2\,\mathrm{TeV} \leq m_{t\bar{t}} \leq 2.2\,\mathrm{TeV}$. Right: same as left for $t\bar{t} + 4$ jets production. No "true" curves are shown, as the statistical uncertainty of the test dataset is too high.

deviations, but within the expected range from the ensemble with its 10 runs. We can also compare the performance of the three architectures. For the standard transformer, we do not observe amplification, corresponding to $G < 1$. The L-GATr generator network does slightly better, with $G \lesssim 1$. Only the LLoCa transformer has a sizeable amplification with $G \gtrsim 1$. All estimated amplification factors are in agreement with the corresponding true values.

In the right panel of Fig. 13 we show the corresponding results for $t\bar{t} + 4$ jets production. Here, the test dataset is smaller, resulting in an integral estimate $2\cdot10^{-4}\pm3\cdot10^{-5}$. This limited precision is not sufficient for a good validation. The averaging amplification estimates suggest that all architectures amplify the training dataset in this phase space region. Comparing their performances, the base transformer is the weakest, although still achieving $G > 1$. The LLoCa transformer and L-GATr both reach significant amplification factors $G \sim 10$.

## 5.2 Differential amplification

As an alternative, we also employ our differential amplification estimate for all three architectures. As described in Section 4, we employ a neural classifier as the summary statistic, trained on all phase-space features [21, 22].

The upper left panel of Fig. 14 shows the scaling for $t\bar{t} + 0$ jets production, comparing generated and training datasets. The standard transformer performs notably worse than the two Lorentz-equivariant architectures. For all three generators, the KS statistic already deviates from its asymptotic behavior before the effective number of generated events matches the number of training events. We estimate $G \approx 0.1$ for the two Lorentz-equivariant networks and $G \approx 0.01$ for the standard transformer. These results are confirmed by the larger test dataset in the upper right panel.

Next, we restrict the phase space to the same $m_{t\bar{t}}$ region as in Section 5.1. We show the results of the KS test compared with the training dataset in the lower left panel of Fig. 14. The Lorentz-equivariant architectures again outperform the standard transformer, and their KS test statistics match the asymptotic band over the full range of generated event numbers. This suggests significant amplification. For a more quantitative picture, we again show the test against the large holdout dataset in the lower right panel, confirming amplification with $G \simeq 2$ for the LLoCa transformer.

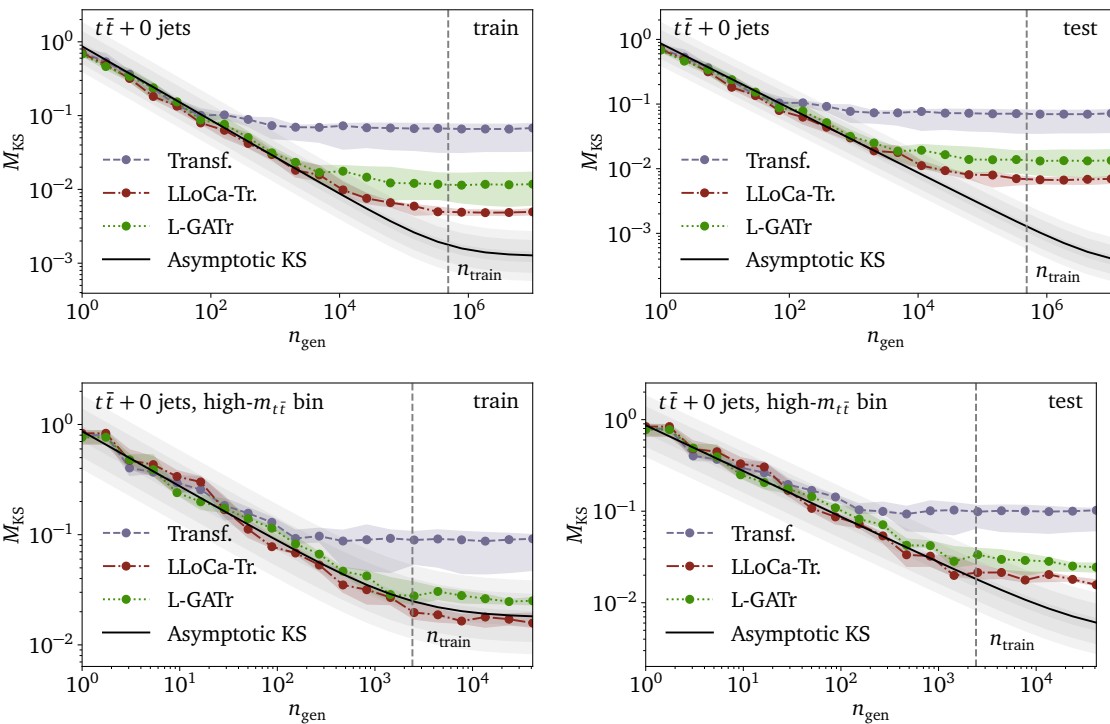

Figure 14: Differential amplification test for the $t\bar{t}+0$ jets events for three different architectures. The two panels on the left compare generated and training events, the two panels on the right use a large holdout testing dataset. The top panels use all events for the hypothesis test, whereas the bottom panels use only events in the high-$m_{t\bar{t}}$ bin defined in Eq. (43). The uncertainties are estimated based on ten independent trainings.

We move on to the $t\bar{t}+4$ jets case in Fig. 15. In the left panels, we see that the three architectures are now significantly closer in their amplification performance, just like for the averaging amplification estimate. While the differential amplification does not indicate any amplification behavior, individual training of both the LLoCa transformer and the L-GATr can reach the asymptotic curve. This suggests that they can probably achieve amplification with a little more fine-tuning.

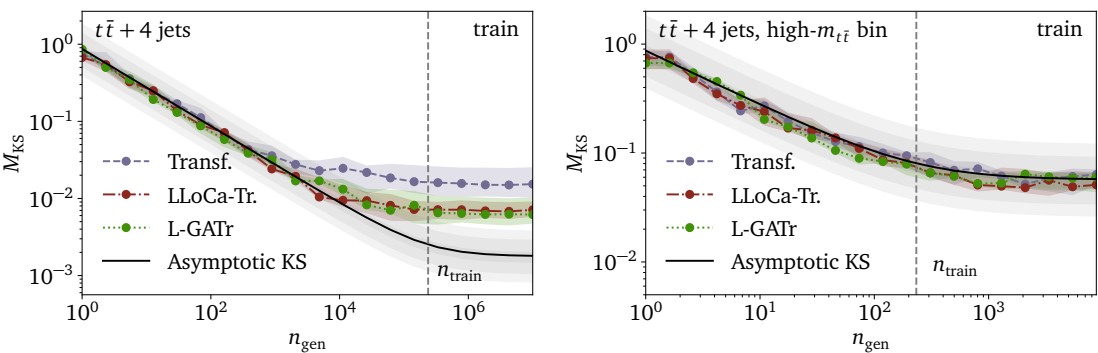

Figure 15: Differential amplification test for the $t\bar{t}+4$ jets events for three different architectures. The left panel uses all events for the hypothesis test, whereas the right panel uses only events in the high-$m_{t\bar{t}}$ bin defined in Eq. (43). No large test dataset is available for this process. The uncertainties are estimated based on ten independent trainings.

As before, we now restrict ourselves to the region $2\,\text{TeV} \leq m_{t\bar{t}} \leq 2.2\,\text{TeV}$, where all three KS test statistics follow the expected asymptotic KS behavior, indicating amplification with $G \approx 5$. Without a large holdout dataset, we cannot quantify the possibly larger amplification factor more precisely.

The amplification factors from averaging and differential estimates differ slightly, which is not unexpected. The averaging estimate is based on a phase space region and has no resolution within that region. Even when we combine regions, the resolution will still be limited by their individual sizes. The differential approach does not have resolution issues, so it can pick up very local differences between the datasets. This will result in smaller amplification factors. Our comparison does not prefer one estimate over the other. Instead, it shows the importance of selecting an amplification measure that is appropriate for a given dataset and application.

# 6  Outlook

The main challenge for LHC physics in the coming years is to prepare for the order-of-magnitude increase in data with the HL-LHC. Generative neural networks are the key to overcome computational obstacles, which means evaluating and controlling their performance is crucial. A fascinating aspect of generative network uncertainties is the amplification factor, the (inverse) ratio of the size of the training dataset and a dataset with statistical power equivalent to the generative network.

We present two complementary approaches to estimate the amplification factor without large hold-out datasets. First, an averaging amplification estimate measures the agreement between the integral of the true likelihood and the generative network likelihood for a specific phase-space volume. We demonstrated how this agreement can be estimated using BNNs or network ensembles. Second, we showed how to employ a KS test based on the likelihood ratio to extract the differential amplification factor, which compares the generated dataset to the available truth datasets differentially without averaging.

As a realistic application, we discussed $t\bar{t} + \text{jets}$ production. Using our amplification measures, we compared three different generative networks: a transformer, L-GATr, and a LLoCa transformer, all three inside a CFM generator. For the averaging amplification test on a high-$m_{t\bar{t}}$ region, we confirmed that the Lorentz-equivariant networks lead to amplification. For the relevant phase space region, we also found evidence for amplification using the more demanding differential amplification definition.

Our study provides a new systematic framework to quantify the statistical amplification of generative networks in LHC physics. It can easily be applied to tasks beyond our toy examples and event generation tasks.

**Code availability**

Our code for generating the toy datasets and evaluating amplification is available at https://github.com/heidelberg-hepml/gan_estimate. The code to reproduce the top pair production datasets is available at https://github.com/heidelberg-hepml/lorentz-frames.

# Acknowledgements

We would like to thank Louis Lyons for asking the right questions, repeatedly, and this way triggering this study and improving the paper. We also thank Benjamin Nachman for useful discussions and feedback. S.D. is supported by the U.S. Department of Energy (DOE), Office of Science under contract DE-AC02-05CH11231. J.S. is funded by the Carl-Zeiss-Stiftung through the project Model-Based AI: Physical Models and Deep Learning for Imaging and Cancer Treatment. N.E. is funded by the Heidelberg IMPRS *Precision Tests of Fundamental Symmetries*. This research is supported through the KISS consortium (05D2022) funded by the German Federal Ministry of Education and Research BMBF in the ErUM-Data action plan, by the Deutsche Forschungsgemeinschaft (DFG, German Research Foundation) under grant 396021762 – TRR 257: *Particle Physics Phenomenology after the Higgs Discovery*, and through Germany's Excellence Strategy EXC 2181/1 – 390900948 (the *Heidelberg STRUC-TURES Excellence Cluster*). Finally, we would like to thank the Baden-Württemberg Stiftung for financing through the program *InternationaleSpitzenforschung*, project *Uncertainties – Teaching AI its Limits* (BWST_ISF2020-010).

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
