# Peer review of "Forecasting Generative Amplification"

_SciPost Physics_

## Round 3 · Referee Report · Humberto Reyes-González (Referee 1) · 2025-12-20

The referee discloses that the following generative AI tools have been used in the preparation of this report:
ChatGPT based on GPT-5.2 was used to
-help organize notes into a clear text
-assist with minor discussions on statistical interpretations.
Strengths
Weaknesses
Report
The construction is mathematically well defined and illustrated with toy models and physics examples. The paper highlights the role of inductive bias and smoothing in generative models and provides a potentially useful performance-based diagnostic. However, several conceptual clarifications are needed to avoid misinterpretation, particularly from a statistical perspective. In this regard, I kindly ask the authors to address the following comments:
-What is meant by 'amplification' should be made explicit. From a purely frequentist point of view, amplification is not possible without introducing additional assumptions, and there is no amplification in the Fisher-information sense. The reported amplification arises entirely from inductive bias (priors, smoothness assumptions, architectural constraints, uncertainty modeling), not from additional information extracted from the data. This is implicit in the manuscript but the distinction should be stated explicitly to avoid misinterpretation.
-It should also be made explicit that the factor G measures performance under a specific, task-dependent metric and within a particular region of phase space. It quantifies variance reduction under a chosen discrepancy measure induced by model assumptions, rather than a general improvement of the learned distribution.
-Section 3.2 effectively probes a single parameter mu; this should be stated explicitly in the interpretation of the results. A simple counterexample where amplification is not obtained would be useful. As a stress test, the authors could repeat this example with very small training samples (e.g. n_train = 10).
-In this context, the interpretation of G would benefit from a discussion of the small-n_train regime, where uncertainty estimates may be prior-dominated and large values of G may arise even when the data weakly constrain the model.
-The statement that the likelihood ratio is the most powerful statistic according to the Neyman–Pearson lemma is strictly true only for known probability density functions. A classifier provides an approximation to a monotonic function of the optimal likelihood ratio, with optimality holding only asymptotically. This distinction should be clarified. Moreover, compressing multivariate data into a one-dimensional score can hide discrepancies present in the full space, and potential dimensionality issues should be mentioned.
-The use of KS tests on classifier outputs has been explored previously (see, e.g., Fig. 14 of arXiv:2305.14137), where more powerful alternatives are discussed. While asymptotic distributions are only known for univariate test statistics, multivariate tests can be calibrated via Neyman constructions (see arXiv:2511.09118, arXiv:2508.02275, arXiv:2409.16336). In particular, arXiv:2511.09118 discusses generative limits due to model mismodeling and is directly relevant here.
-All of the above mentioned references should be cited.
-Exact values of G should be quoted in the text, including cases where (G < 1).
Recommendation
Ask for minor revision

---

## Editorial Decision

unknown